# CoIn: Coverage and Informativeness-Guided Token Reduction for Efficient Large Multimodal Models

## Abstract

Large Multimodal Models (LMMs) have shown remarkable success in image understanding tasks. LMMs encode visual and textual inputs into tokens, which are then fed into Large Language Models (LLMs). However, the large number of visual tokens poses a major bottleneck for inference efficiency and memory usage. Reducing visual tokens is a promising training-free solution, but existing methods remain limited: importance-based approaches often yield redundant selections, diversity-based ones overlook differences among tokens themselves. Two-stage hybrid methods inherit shortcomings form importance-based selection and result in suboptimal choices. To address this, we formulate token reduction as an optimal subset selection problem and identify two key criteria for a good subset: informativeness and coverage, to guide the selection that best preserves LLM output fidelity. Based on these principles, we propose **CoIn**, a token selection framework that jointly optimizes both. CoIn integrates visual saliency, cross-modal relevance, and representational novelty into a unified scoring function, enabling the selection of a compact yet expressive token subset. It is efficient, model-agnostic, and compatible with modern inference accelerators. Experiments on multiple benchmarks demonstrate that CoIn substantially reduces computation and memory cost while maintaining strong task performance.

## 1 Introduction

In recent years, Large Language Models (LLMs) (Touvron et al., 2023; Achiam et al., 2023; Bai et al., 2023) have transformed natural language processing. Building on this success, Large Multimodal Models (LMMs) (Liu et al., 2023; 2024a; Li et al., 2024) have emerged as a powerful extension that integrates visual and linguistic modalities for unified multimodal reasoning. By aligning vision encoders (*e.g.* CLIP (Radford et al., 2021)) with language backbones, LMMs excel in tasks like multimodal reasoning (Wang et al., 2024), and visual question answering (VQA).

Despite these advances, LMMs face a critical computational bottleneck. They typically encode input images into a sequence of visual tokens that are then concatenated with text tokens for LLM processing. The number of visual tokens grows with image resolution, leading to long input sequences. Since LLM inference time and memory cost scale quadratically with sequence length (Dao et al., 2022; Choromanski et al., 2020), this design leads to substantial latency and memory usage, hindering the practical deployment of LMMs in real-world scenarios such as mobile devices and chat assistants.

To alleviate this, previous works (Chen et al., 2024; Shang et al., 2024) have proposed various token reduction techniques, which can be broadly divided into three categories. **Importance-based** methods estimate token saliency using attention weights (Xing et al., 2024; Zhang et al., 2024b) or [CLS]-similarity scores (Shang et al., 2024), retaining those with the highest individual scores. While effective in preserving key details, they often yield redundant selections with low information density. Moreover, their reliance on attention can introduce bias (Wen et al., 2025) and conflict with efficient mechanisms like FlashAttention (Dao et al., 2022), while [CLS]-guided scoring limits generalizability across vision encoders. **Diversity-based** methods (Alvar et al., 2025; Jeddi et al., 2025) instead select a representative subset through clustering or pairwise similarity. However,

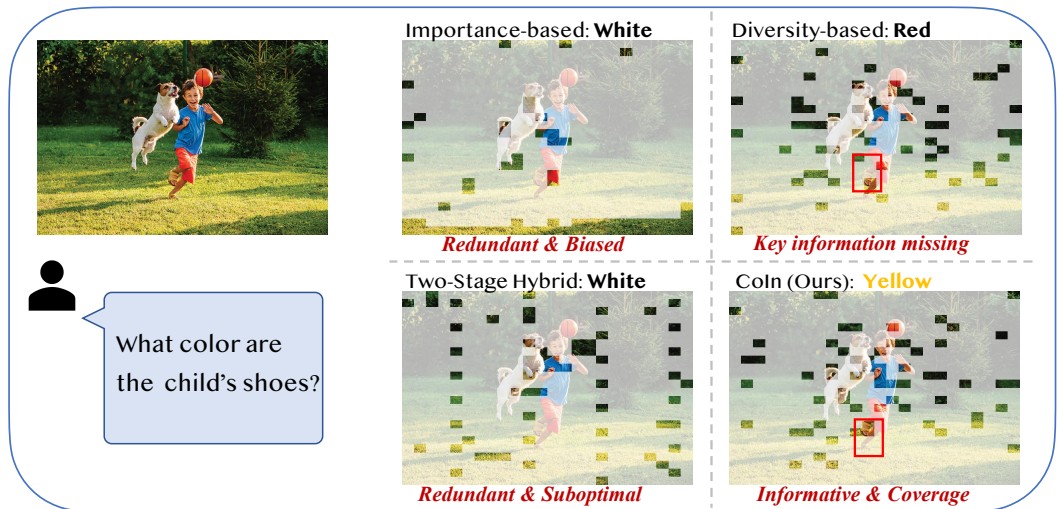

Figure 1: Comparison between baselines and CoIn.

they implicitly treat all tokens as equally important, overlooking intrinsic saliency and semantic relevance, which leads to missing critical information. **Hybrid** methods (Yang et al., 2025; Shang et al., 2024) attempt to combine both paradigms in a two-stage manner: first selecting salient tokens via importance scoring, then applying token merging in the remained tokens. While this mitigates some bias, it inherits the limitations of importance-based selection and fails to effectively reduce redundancy. As illustrated in Figure 1, these shortcomings hinder existing approaches from selecting an optimal token subset, ultimately degrading LMM responses.

To overcome these limitations, we rethink token reduction as a unified *optimal subset selection* problem driven by two complementary criteria: **informativeness** and **coverage**. First, to address the redundancy and bias issues in importance-based approaches, we define *informativeness* as a token's joint contribution to model prediction, integrating its intrinsic visual saliency and semantic alignment with the accompanying text. Second, instead of patching importance-based selection with a separate token merging step as in two-stage hybrids, we introduce *coverage* as a diversity-aware criterion and integrate it directly into a single joint optimization process, ensuring preservation of novel visual content. Building on these insights, we propose **CoIn**, a training-free framework that simultaneously optimizes for informativeness and coverage, producing a compact token subset that is both individually salient and collectively representative.

Extensive experiments on multiple benchmarks demonstrate that CoIn significantly reduces computation and memory costs while maintaining strong downstream task performance. For example, when applied to LLaVA-1.5-7B model, CoIn achieves an average accuracy of 91.29% across 9 benchmarks with 94.4% visual tokens reduced. Beyond performance, CoIn offers practical deployment advantages: it is fully compatible with efficient inference techniques such as KV caching and FlashAttention, and, by avoiding [CLS]-token dependency, remains model-agnostic across diverse vision encoders.

To summarize, our main contributions are threefold:

- We recast token reduction as an *optimal subset selection* problem and introduce two principled criteria—**informativeness** and **coverage**—to jointly ensure token saliency and diversity, thereby preserving the fidelity of the LMM output.

- We develop **CoIn**, a training-free framework that unifies visual saliency, cross-modal semantic alignment, and representational diversity in a single joint selection process, overcoming the bias, redundancy, and disjoint design issues of prior approaches.

- Through extensive experiments on diverse benchmarks and reduction ratios, we show that CoIn achieves substantial token reduction (up to 94.4%) with minimal performance loss, consistently outperforming state-of-the-art methods.

## 2 RELATED WORK

### 2.1 LARGE MULTIMODAL MODELS

Recent advances in large multimodal models (LMMs) (Cheng et al., 2024; Team et al., 2024; Achiam et al., 2023) have demonstrated strong performance across vision-language tasks by combining pretrained vision encoders with large language models. These models, such as LLaVA (Liu et al., 2023; 2024a), typically convert visual inputs into token sequences and feed them into LLMs for unified processing. Specifically, LLaVA-1.5 (Liu et al., 2023) encodes a $336\times336$ image into 576 visual tokens, while LLaVA-NeXT (Liu et al., 2024a) supports higher-resolution and dynamic input, producing up to 2,880 visual tokens per image. As the number of visual tokens increases significantly with resolution or input length, it introduces substantial computational overhead, motivating recent efforts to improve token efficiency and inference speed.

### 2.2 VISUAL TOKEN REDUCTION

A prominent line of work to improve the efficiency of LMMs focuses on reducing the number of visual tokens. Existing methods can be broadly categorized into three types. The first type estimates the importance of each visual token individually. One common approach relies on attention scores within the language model to identify less important tokens (Chen et al., 2024; Ye et al., 2025). However, these scores often suffer from attention bias (Wen et al., 2025), and are incompatible with efficient attention mechanisms (Dao et al., 2022). An alternative strategy leverages encoder-side signals, such as [CLS] token, to guide token selection (Shang et al., 2024). While effective in certain settings, such methods typically depend on specific encoder architectures, limiting their general applicability across different LMM frameworks. The second type emphasizes diversity, aiming to eliminate redundant tokens based on feature similarity (Jeddi et al., 2025; Alvar et al., 2025). These approaches treat all visual tokens equally, without considering differences among tokens themselves, which can result in the omission of critical information. The third category attempts to combine importance and diversity in a two-stage manner. PruMerge+(Shang et al., 2024) first uses [CLS] token to select important tokens, followed by clustering to retrieve complementary tokens. Similarly, VisionZip (Yang et al., 2025) first selects tokens based on attention weights, and then merges remaining ones based on similarity. While this mitigates some bias, it inherits the limitations of importance-based selection and fails to effectively reduce redundancy.

## 3 METHODOLOGY

### 3.1 PRELIMINARY

Large Multimodal Models (LMMs) typically consist of four main components: a vision encoder $E_v$, a projector $P$, a text encoder $E_t$, and an LLM $f_\phi$. These modules work together to enable the fusion and reasoning over visual and textual information. Given an input image $x_v$, the vision encoder first extracts visual features, which are then mapped into the LLM-compatible embedding space via the projector, resulting in visual embeddings $V = P(E_v(x_v))$. Meanwhile, the text input $x_t$ is encoded by the Text Encoder into language tokens $T = E_t(x_t)$. The projected visual embeddings $V$ are then concatenated with the text tokens $T$ to form an input sequence $[V; T]$, which is fed into LLM to perform downstream tasks:

$$y = f\phi([V; T]) \tag{1}$$

Visual token reduction accelerates LMMs and lowers computational overhead by reducing the number of visual tokens. Given a projected visual token set $V = P(E_v(x_v)) = \{v_1, ..., v_N\}$, our goal is to select a subset $V' \subseteq V$ of size $K$ ($K \ll N$) such that the LLM output remains close to that with full token set. Formally, we define the token selection objective as:

$$V' = \underset{V' \subseteq V, |V'|=K}{\arg\min} \mathcal{D}(f_\phi([V; T]), f_\phi([V'; T])), \tag{2}$$

where $\mathcal{D}(\cdot, \cdot)$ denotes LLM output difference.

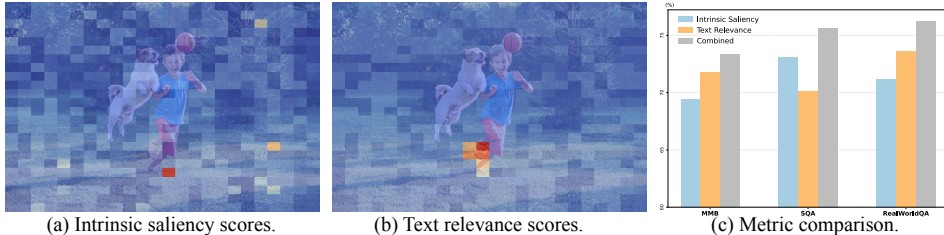

(a) Intrinsic saliency scores.     (b) Text relevance scores.     (c) Metric comparison.

Figure 2: Illustration of informativeness criteria: (a) and (b) visualize token scores for the prompt "What color are the child's shoes?", based on intrinsic saliency and text relevance respectively. Red indicates high scores, while blue denotes low scores. (c) reports performance of using visual saliency only, text relevance only, and their combination.

## 3.2 DESIDERATA FOR A GOOD TOKEN SUBSET

Selecting a subset of visual tokens that faithfully preserves the LMMs output is non-trivial. Unlike textual inputs, where syntactic cues and token order offer strong priors for importance estimation, visual tokens derived from image patches lack such explicit structures. Therefore, identifying which tokens to retain must be guided by principles that maintain the fidelity of the original input. Toward this goal, we identify two key desiderata:

**Informativeness.** This criterion refers to the degree that a visual token contributes to the model's prediction. A token's informativeness stems from two sources: its intrinsic saliency within the visual modality, and its alignment with the accompanying text. Tokens that are visually prominent or semantically correlated with the text are more likely to play a pivotal role in the reasoning process. To highlight the complementary nature of these two factors, we visualize token scores when using only intrinsic saliency or only text relevance (Figure 2(a) and 2(b)), demonstrating that each criterion emphasizes different regions. We further evaluate three selection strategies: relying solely on intrinsic saliency, solely on text relevance, and their combination. As reported in Figure 2(c), the combined consistently achieves superior results, indicating that neither criterion alone is sufficient to capture all informative tokens.

**Coverage.** In addition to individual informativeness, a high-quality token subset should offer comprehensive coverage. Visual inputs often contain redundancy (e.g., multiple similar patches in the background), thus selecting only the most informative tokens may lead to overrepresentation of those regions. The notion of coverage encourages the inclusion of tokens from distinct regions or object parts, thereby preserving the holistic structure and context of the scene.

Taken together, informativeness and coverage ensure that the selected tokens are both informative and representative, forming a minimal yet effective input for the LMMs.

## 3.3 THE DESIGN OF COIN

In this section, we instantiate an efficient algorithm for selecting a subset of visual tokens that jointly satisfies the desiderata of informativeness and coverage, as discussed above. Our formulation can be viewed as a greedy procedure for constructing a sparse and expressive subspace in the joint visual-linguistic representation space. At the core of our method lies a composite scoring function that integrates token-wise informativeness with global subspace diversification.

**Scoring Function.** Given the complete visual token set $V = \{v_1, \ldots, v_N\}$ and the text token set $T = \{t_1, \ldots, t_M\}$, we define a scoring function $\text{Score}(v_i)$ that evaluates the desirability of each token $v_i$ based on the following two terms:

*Informativeness Score.* We quantify the informativeness of each visual token $v_i$ based on both visual saliency and cross-modal alignment:

$$\text{Info}(v_i) = \beta \cdot \|v_i\|_p + (1 - \beta) \cdot \frac{1}{M} \sum_{j=1}^{M} \cos(v_i, t_j), \quad (3)$$

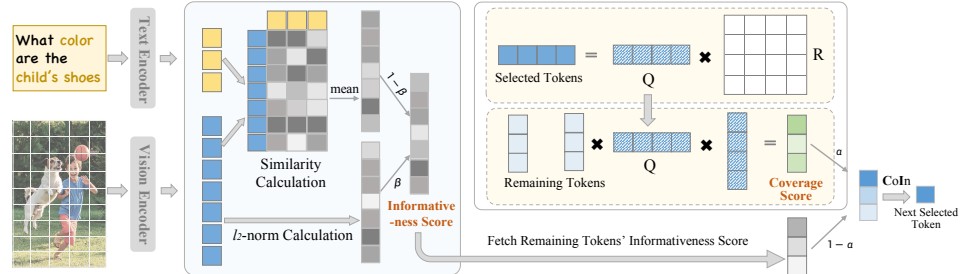

Figure 3: Overview of our proposed method (CoIn), which iteratively selects the visual tokens with the highest combined score.

where $\|v_i\|_p$ is the $\ell_p$-norm of the token (default $p = 2$), serving as a proxy for visual saliency, while the cosine similarity with each text token $t_j$ measures cross-modal relevance. The hyperparameter $\beta \in [0, 1]$ governs the trade-off between visual and textual cues.

*Coverage Score.* Inspired from volume-based subset selection methods (Deshpande & Rademacher, 2010), we define coverage as the degree of representational novelty, *i.e.*, how much new information a token contributes relative to the current selected subset. This is measured via orthogonal projection in the feature space. Let $S \subset V$ be the currently selected token set, and $\mathbf{X}_S$ denote the matrix of their embeddings. Performing a QR decomposition (Gander, 1980) on $\mathbf{X}_S$ yields:

$$\mathbf{X}_S = \mathbf{QR}, \tag{4}$$

where the columns of $\mathbf{Q}$ form an orthonormal basis spanning the subspace of $S$. For any remaining token $v_i \in V \setminus S$, its coverage score is defined as the $\ell_2$ norm of its residual after projection onto this subspace:

$$\text{Cov}(v_i) = \left\| v_i - \mathbf{Q}\mathbf{Q}^\top v_i \right\|_2 . \tag{5}$$

This residual reflects how much additional information $v_i$ provides beyond what is already captured by $S$, thereby quantifying its contribution to the overall representational diversity.

*Combined Score.* The final score used for selection is a convex combination of informativeness and coverage:

$$\text{Score}(v_i) = \alpha \cdot \text{Cov}(v_i) + (1 - \alpha) \cdot \text{Info}(v_i), \tag{6}$$

where $\alpha \in [0, 1]$ balances the emphasis between preserving informativeness and encouraging diversity.

**Selection Strategy.** Starting from an empty set, our algorithm proceeds in a greedy manner. We firstly select the token with the highest informativeness score. At each subsequent iteration, we select the token with the highest combined score. After each selection, the coverage component is incrementally updated to reflect the expanded token span. This process continues until a budget of $K$ tokens is reached.

## 4 EXPERIMENT

### 4.1 EXPERIMENTAL SETTINGS

**Models and Baselines.** We conduct experiments on several popular MLLMs, including LLaVA-1.5-7B (Liu et al., 2023), LLaVA-1.5-13B (Liu et al., 2023), and LLaVA-NeXT-7B (Liu et al., 2024a), to demonstrate the generality of our approach. All tested MLLMs use the CLIP vision encoder (Radford et al., 2021). We consider four training-free baselines: importance-based method PDrop (Xing et al., 2024), two-stage hybrid methods like PruMerge+ (Shang et al., 2024), VisionZip (Yang et al., 2025), and diversity-based method DivPrune (Alvar et al., 2025).

Table 1: Performance on LLaVA-1.5-7B. "Avg." indicates average performance relative to original model across 9 benchmarks.

| Method | GQA | MMB | MME | POPE | VQA$^{\text{Text}}$ | VizWiz | OCRB | SQA$^{\text{IMG}}$ | RWQA | Avg. |
|---|---|---|---|---|---|---|---|---|---|---|
| *Upper Bound, 576 Tokens* | | | | | | | | | | |
| LLaVA-1.5-7B | 62.0 | 64.1 | 1508 | 85.9 | 46.1 | 54.3 | 0.31 | 69.5 | 55.8 | 100% |
| *Retained 128 tokens (↓77.8%)* | | | | | | | | | | |
| PDrop (CVPR25) | 57.1 | 61.7 | 1445 | 77.4 | 43.9 | 53.7 | 0.29 | 69.0 | 51.1 | 94.70% |
| PruMerge+ (ICCV25) | 57.6 | 60.1 | 1381 | 81.0 | 39.2 | 56.0 | 0.28 | 69.5 | 49.9 | 93.25% |
| VisionZip (CVPR25) | 57.6 | 62.2 | 1445 | 82.9 | 43.6 | 54.1 | 0.30 | 68.6 | 51.9 | 95.93% |
| DivPrune (CVPR25) | 59.2 | 62.3 | 1403 | 86.6 | 42.0 | 56.4 | 0.29 | 68.6 | 49.7 | 95.69% |
| CoIn ($\alpha=0.9, \beta=0.6$) | 59.3 | 62.4 | 1406 | 87.3 | 43.1 | 56.0 | 0.30 | 69.2 | 50.7 | **96.75%** |
| *Retained 64 tokens (↓88.9%)* | | | | | | | | | | |
| PDrop (CVPR25) | 46.3 | 48.1 | 984.3 | 41.3 | 39.6 | 50.4 | 0.27 | 68.7 | 49.3 | 79.45% |
| PruMerge+ (ICCV25) | 55.1 | 58.7 | 1295 | 75.5 | 37.7 | 56.7 | 0.27 | 69.5 | 48.2 | 90.19% |
| VisionZip (CVPR25) | 55.2 | 60.1 | 1373 | 77.0 | 42.0 | 54.7 | 0.28 | 68.9 | 50.9 | 92.76% |
| DivPrune (CVPR25) | 57.6 | 59.5 | 1348 | 85.8 | 39.1 | 57.5 | 0.27 | 68.0 | 49.2 | 93.14% |
| CoIn ($\alpha=0.9, \beta=0.7$) | 57.8 | 59.8 | 1378 | 86.2 | 41.0 | 57.6 | 0.28 | 68.2 | 49.8 | **94.49%** |
| *Retained 32 tokens (↓94.4%)* | | | | | | | | | | |
| PruMerge+ (ICCV25) | 52.6 | 55.0 | 1202 | 70.4 | 33.3 | 56.7 | 0.24 | 68.8 | 45.1 | 84.93% |
| VisionZip (CVPR25) | 51.7 | 57.0 | 1250 | 68.8 | 36.9 | 55.3 | 0.25 | 68.2 | 48.1 | 86.86% |
| DivPrune (CVPR25) | 54.6 | 57.6 | 1268 | 81.2 | 34.9 | 56.8 | 0.25 | 67.6 | 47.2 | 88.87% |
| CoIn ($\alpha=0.9, \beta=0.8$) | 55.7 | 58.3 | 1326 | 84.0 | 37.4 | 57.5 | 0.26 | 69.0 | 48.2 | **91.29%** |

**Datasets, Tasks and Metrics.** We select a diverse set of representative multimodal understanding and reasoning tasks, covering both image-text question answering and complex inference capabilities. Specifically, we evaluate on the following 9 datasets and benchmarks: GQA (Hudson & Manning, 2019), MMBench (Liu et al., 2024b), MME (Fu et al., 2023), POPE (Li et al., 2023), TextVQA (Singh et al., 2019), VizWiz (Gurari et al., 2018), OCRBench (Liu et al., 2024c), ScienceQA-IMG (SQA) (Lu et al., 2022), and RealWorldQA (rea, 2025). These datasets span various task types, including multiple-choice question answering, open-ended question answering, and comprehensive multimodal understanding involving both visual and textual inputs.

For performance evaluation, we adopt standard metrics based on the nature of each task, including Accuracy, Exact Match (EM), F1 Score, Perception Score (P-score) (Fu et al., 2023) for QA tasks. In all evaluation metrics reported in this work, higher values indicate better task performance. Please refer to the appendix for additional details. For time and memory usage, lower values reflect better efficiency.

**Implementation Details.** All experiments are conducted on 4 NVIDIA A800 80GB GPUs using the lmms-evals package (Zhang et al., 2024a) for benchmarking all models and baselines. Results are reported with a batch size of 1.

## 4.2 MAIN RESULTS

We first evaluate our method on LLaVA-1.5-7B and LLaVA-1.5-13B across nine diverse multimodal benchmarks under varying levels of visual token retention. These benchmarks cover a wide spectrum of tasks, including general-purpose VQA (GQA, TextVQA, VizWiz), OCR-heavy datasets (OCR-Bench), commonsense reasoning (ScienceQA-img, RealWorldQA), hallucination detection (POPE), and comprehensive instruction-following tests (MME, MMBench).

As shown in Table 1 and Table 2, we compare our method against several training-free baselines, under 3 visual token budgets: 128, 64, and 32 (corresponding to 77.8%, 88.9%, and 94.4% pruning rates from the full 576 tokens).

Table 2: Performance on LLaVA-1.5-13B. "Avg." indicates average performance relative to original model across 9 benchmarks.

| Method | GQA | MMB | MME | POPE | VQA$^{\text{Text}}$ | VizWiz | OCRB | SQA$^{\text{IMG}}$ | RWQA | Avg. |
|---|---|---|---|---|---|---|---|---|---|---|
| *Upper Bound, 576 Tokens* | | | | | | | | | | |
| LLaVA-1.5-13B | 63.3 | 68.9 | 1523 | 85.9 | 48.7 | 56.6 | 0.34 | 72.8 | 55.0 | 100.00% |
| *Retained 128 tokens (↓77.8%)* | | | | | | | | | | |
| PDrop (CVPR25) | 60.0 | 65.2 | 1476 | 85.4 | 42.5 | 56.7 | 0.30 | 73.2 | 50.3 | 94.99% |
| PruMerge+ (ICCV25) | 57.4 | 64.3 | 1400 | 80.8 | 41.2 | 54.9 | 0.29 | 73.0 | 50.1 | 92.18% |
| VisionZip (CVPR25) | 57.8 | 65.2 | 1439 | 82.2 | 45.4 | 54.6 | 0.32 | 73.1 | 50.2 | 94.79% |
| DivPrune (CVPR25) | 58.8 | 65.8 | 1450 | 86.2 | 43.2 | 56.5 | 0.31 | 72.8 | 50.0 | 94.89% |
| CoIn ($\alpha$=0.9, $\beta$=0.6) | 59.2 | 66.2 | 1466 | 87.0 | 44.6 | 56.6 | 0.32 | 73.4 | 50.9 | **96.22%** |
| *Retained 64 tokens(↓88.9%)* | | | | | | | | | | |
| PDrop (CVPR25) | 54.1 | 63.4 | 1240 | 66.1 | 39.0 | 53.2 | 0.28 | 69.1 | 49.0 | 86.27% |
| PruMerge+ (ICCV25) | 55.6 | 62.3 | 1316 | 74.1 | 39.5 | 55.8 | 0.29 | 72.6 | 49.4 | 89.45% |
| VisionZip (CVPR25) | 56.0 | 63.7 | 1402 | 76.0 | 41.8 | 56.0 | 0.31 | 73.2 | 50.1 | 92.21% |
| DivPrune (CVPR25) | 57.4 | 63.8 | 1486 | 85.0 | 40.9 | 58.3 | 0.30 | 71.5 | 48.5 | 93.48% |
| CoIn ($\alpha$=0.9, $\beta$=0.6) | 58.0 | 64.4 | 1482 | 86.1 | 42.2 | 58.0 | 0.30 | 72.0 | 50.6 | **94.57%** |
| *Retained 32 tokens (↓94.4%)* | | | | | | | | | | |
| PruMerge+ (ICCV25) | 54.2 | 60.0 | 1253 | 68.0 | 34.4 | 55.3 | 0.26 | 71.3 | 47.7 | 84.92% |
| VisionZip (CVPR25) | 52.8 | 61.6 | 1281 | 67.1 | 37.3 | 56.8 | 0.27 | 72.0 | 48.9 | 86.59% |
| DivPrune (CVPR25) | 55.5 | 61.3 | 1390 | 75.7 | 35.1 | 57.8 | 0.27 | 70.8 | 49.1 | 88.58% |
| CoIn ($\alpha$=0.8, $\beta$=0.9) | 56.5 | 62.0 | 1402 | 82.2 | 37.4 | 58.3 | 0.28 | 71.5 | 49.8 | **91.04%** |

**Performance on LLaVA-1.5-7B.** When retaining 128 tokens, our method achieves **96.75%** of the original full-token performance, surpassing the best-performing baseline (DivPrune). As the retention drops to 64, the gap between methods becomes more pronounced. PDrop suffer noticeable degradation, losing over **20%** of their original score. In contrast, our method maintains a robust **94.49%** of the original performance, outperforming VisionZip and DivPrune by **1.73%** and **1.35%**, respectively. This confirms the advantage of jointly considering both token importance and diversity during selection. In the extreme case of only 32 tokens retained (*i.e.*, 94.4% pruned), most baselines experience a drastic performance collapse due to the loss of essential semantic information and excessive redundancy in retained tokens. Our method, however, still preserves **91.29%** of the original score, significantly outperforming DivPrune by **2.42%**. We observe particularly strong results on hallucination-sensitive tasks such as POPE and real-world QA datasets, indicating our method's ability to retain critical visual cues under aggressive pruning.

**Performance on LLaVA-1.5-13B.** Scaling to the larger LLaVA-1.5-13B, we observe a consistent performance pattern. Our approach achieves **96.22%**, **94.57%**, and **91.04%** retention rates at 128, 64, and 32 token settings, respectively—again outperforming all baselines by a substantial margin. Notably, while LLaVA-13B tends to be more sensitive to visual information loss due to its higher capacity and richer vision-language alignment, our method maintains stable performance, showcasing its generalizability across model scales. We also note strong performance on POPE, where hallucination control is critical. Our method's ability to preserve factual grounding further underscores the semantic faithfulness of the retained token subset. Additionally, tasks such as MMBench and ScienceQA—which require fine-grained visual reasoning—also see minimal drops, indicating our approach's ability to retain not just the most salient but also diverse and complementary visual evidence.

These results collectively demonstrate that our joint coverage-informativeness pruning strategy is highly effective across varying compression levels and model sizes. It avoids over-selecting salient but redundant tokens and preserves a balanced representation of the visual scene. As demonstrated on hallucination-sensitive tasks like POPE, our method mitigates common failure cases arising from incomplete or biased visual grounding.

Table 3: Performance on LLaVA-NEXT-7B. "Avg." indicates average performance relative to original model across 9 benchmarks.

| Method | GQA | MMB | MME | POPE | VQA$^{Text}$ | VizWiz | OCRB | SQA$^{IMG}$ | RWQA | Avg. |
|---|---|---|---|---|---|---|---|---|---|---|
| *Upper Bound, 2880 Tokens* | | | | | | | | | | |
| LLaVA-NeXT-7B | 64.2 | 67.1 | 1519 | 86.5 | 64.9 | 60.8 | 0.52 | 70.4 | 57.7 | 100% |
| *Retained 640 tokens (↓77.8%)* | | | | | | | | | | |
| PDrop (CVPR25) | 58.6 | 63.4 | 1471 | 81.8 | 56.3 | 53.3 | 0.37 | 65.9 | 50.7 | 89.27% |
| VisonZip (CVPR25) | 59.7 | 64.2 | 1474 | 83.3 | 58.9 | 57.5 | 0.40 | 67.9 | 53.9 | 92.70% |
| DivPrune (CVPR25) | 61.3 | 64.2 | 1467 | 85.9 | 54.7 | 58.7 | 0.37 | 67.6 | 52.4 | 91.78% |
| CoIn ($\alpha=0.7, \beta=0.9$) | 61.5 | 65.1 | 1500 | 86.3 | 58.4 | 58.4 | 0.41 | 67.8 | 54.12 | **93.97%** |
| *Retained 320 tokens (↓88.9%)* | | | | | | | | | | |
| VisonZip (CVPR25) | 58.4 | 62.7 | 1404 | 80.1 | 55.5 | 55.5 | 0.34 | 68.2 | 51.0 | 88.55% |
| DivPrune (CVPR25) | 59.7 | 64.1 | 1410 | 83.4 | 49.5 | 57.3 | 0.32 | 67.3 | 49.7 | 87.92% |
| CoIn ($\alpha=0.9, \beta=0.8$) | 59.9 | 63.8 | 1449 | 85.4 | 53.1 | 57.6 | 0.34 | 68.1 | 51.5 | **89.92%** |
| *Retained 160 tokens (↓94.4%)* | | | | | | | | | | |
| VisonZip (CVPR25) | 56.3 | 59.5 | 1334 | 74.6 | 48.0 | 55.5 | 0.29 | 68.0 | 48.1 | 83.7% |
| DivPrune (CVPR25) | 57.8 | 61.5 | 1354 | 79.8 | 45.2 | 57.8 | 0.27 | 67.7 | 49.3 | 84.53% |
| CoIn ($\alpha=0.9, \beta=0.7$) | 58.1 | 63.1 | 1369 | 82.2 | 49.2 | 57.5 | 0.30 | 68.0 | 49.4 | **86.55%** |

## 4.3 PUSHING TO HIGHER RESOLUTION

In this section, we apply our method to LLaVA-Next-7B, a large multimodal model capable of handling high-resolution images with up to 2880 visual tokens. As the number of tokens increases, the inference cost of the model rises significantly. We compare against PDro, VisionZip, and DivPrune under various token retention settings (640, 320, and 160), as shown in Table3,

At 320 tokens, most methods experience moderate drops in performance. While DivPrune drops to **87.92%**, our method maintains **89.92%**. Even under the extreme setting of 160 tokens (94.4% pruned), our method still preserves **86.55%**, outperforming DivPrune by **2.02%** and VisionZip by **2.86%**. Across all reduction levels, our method consistently outperforms prior baselines, demonstrating strong robustness on high-resolution model LLaVA-Next. In particular, our method maintains better balance across both reasoning-oriented tasks (e.g., GQA, MME) and OCR-heavy benchmarks (e.g., TextVQA, OCRBench), highlighting the benefit of our joint importance-diversity selection strategy.

## 4.4 EFFICIENCY ANALYSIS

In this section, we analyze the efficiency of our proposed method from 3 perspectives: GPU memory usage, prefill time, and decode time. The experiment is conducted on the LLaVA-NEXT-7B model using the POPE dataset on a single NVIDIA A100-80GB GPU. The results are summarized in Table 4, where we compare our method against the original model as well as PDrop, PruneMerge, VisionZip, and Di-

Table 4: Efficiency Analysis.

| Method | Max GPU Mem (GB) | Prefill Time | Decoding Time | Score F1 |
|---|---|---|---|---|
| Original | 16.8 | 233ms | 27ms | 86.5 |
| VisionZip | 14.9 | 36ms | 21ms | 80.1 |
| DivPrune | **13.9** | 36ms | 21ms | 83.4 |
| CoIn | 14.1 | 36ms | 21ms | **85.4** |

vPrune the number of visual tokens is reduced from 2880 to 320. In terms of CUDA latency, our method reduces the prefill and decode time by 6.5× and 1.3× respectively. Beyond runtime latency, our method also lowers GPU memory consumption.

## 4.5 ABLATION STUDIES

To better understand the contribution of each component in our visual token selection framework, we conduct a series of ablation studies using the LLaVA-1.5-7B backbone. Specifically, we analyze

Table 5: Ablation study on informativeness (Info) and coverage (Cov).

| Variants | Cov | Info | POPE | GQA | MME |
|---|---|---|---|---|---|
| Original | - | - | 85.9 | 62.0 | 1508 |
| (i) | ✗ | ✓ | 73.7 | 51.2 | 1288 |
| (ii) | ✓ | ✗ | 74.4 | 53.0 | 1307 |
| CoIn | ✓ | ✓ | **86.2** | **57.8** | **1378** |

Table 6: Ablation of informativeness term. IS: intrinsic saliency; TR: text relevance.

| Variants | IS | TR | VizWiz | RealWorldQA |
|---|---|---|---|---|
| Original | - | - | 54.3 | 55.8 |
| (i) | ✓ | ✗ | 51.4 | 39.7 |
| (ii) | ✗ | ✓ | 51.1 | 41.1 |
| Combination | ✓ | ✓ | **51.8** | **42.6** |

the impact of (1) combining coverage and informativeness, (2) the decomposition of informativeness into intrinsic saliency and text relevance, and (3) different weighting strategies for $\alpha$ and $\beta$.

**Impact of Combining Coverage and Informativeness.** We conduct an ablation study to evaluate the importance of the two components in our scoring function: informativeness and coverage. With 64 visual tokens retained, we compare CoIn with two simplified variants: (i) informativeness-only, and (ii) coverage-only. As shown in Table 5, both variants lead to clear performance drops compared to the full components. The informativeness-only variant performs the worst, likely due to redundancy among selected tokens. The coverage-only variant preserves diversity but lacks focus on informative content, resulting in suboptimal performance. In contrast, our full strategy, which balances both aspects, achieves the best overall results across benchmarks. Notably, it even surpasses the original unpruned model on POPE, showing that compact yet well-chosen tokens can improve grounding performance while reducing computational cost.

**Effect of Intrinsic Saliency and Text Relevance.** We conduct an ablation study to isolate the contributions of the two components of the informativeness term, while disabling coverage by setting $\alpha=0$ (no coverage). We compare three settings: (i) using only intrinsic visual saliency (IS), (ii) using only text-visual semantic relevance (TR), and (iii) combining both. In all cases, we keep 32 visual tokens based on the chosen variants. As shown in Table 6, using only intrinsic saliency or text relevance yields similar performance on VizWiz, suggesting both visual and semantic signals are useful for perception-focused tasks. However, on RealWorldQA, which requires strong language grounding, text relevance outperforms saliency by a clear margin. Notably, combining both components leads to consistent improvements across both benchmarks, highlighting their complementary strengths: saliency brings spatial awareness, while text relevance ensures semantic alignment. These results validate our design of jointly modeling both factors for more effective token selection.

**Choice of $\alpha$ and $\beta$.** We investigate the impact of different $\alpha$ and $\beta$ values on task performance, as shown in Figure 4. For a fixed $\beta = 0.9$, we vary $\alpha$ in $\{0.5, 0.6, 0.7, 0.8, 0.9\}$. For a fixed $\alpha = 0.9$, we vary $\beta$ from 0 to 1, and report the performance. The results suggest that different tasks exhibit varying preferences for intrinsic saliency, text relevance, and coverage. In practical applications, these parameters can be fine-tuned to optimize performance for specific scenarios.

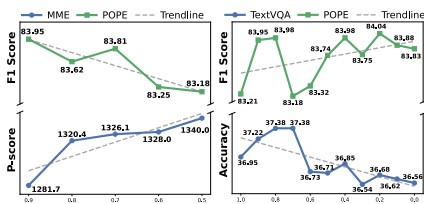

Figure 4: Left: varying $\alpha$ with $\beta=0.9$; Right: varying $\beta$ with $\alpha=0.9$.

## 5 CONCLUSION

In this work, we reformulated the token reduction problem for large multimodal models as an optimal subset selection task. We introduced two key criteria: *informativeness* and *coverage*, and proposed **CoIn**, a training-free framework that jointly optimizes these objectives to produce compact yet representative token subsets. Extensive experiments across multiple benchmarks demonstrate that CoIn achieves substantial token reduction with minimal performance degradation. These results position CoIn as an effective approach for improving the efficiency and scalability of LMMs.

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

## A  THE USE OF LARGE LANGUAGE MODELS

Large Language Models (LLMs) were only used to assist with minor language polishing and formatting. No LLMs were used for data analysis, experiment design, or generation of original content. All conceptualization, modeling, implementation, and evaluation were performed by the authors.

## B  ETHICS STATEMENT

This work does not involve human subjects, personally identifiable information, or sensitive data. No proprietary or non-public datasets are used. The proposed methods and findings do not present foreseeable harmful insights, applications, or security/privacy risks. There are no conflicts of interest, sponsorships, or legal/ethical concerns associated with this study.

## C  REPRODUCIBILITY STATEMENT

We have taken several steps to ensure the reproducibility of our results. The experimental settings and hyperparameter configurations are described in detail in Section 4.1. Complete mathematical algorithm is provided in Appendix D.1. The source code will be submitted as supplementary material and released publicly upon acceptance. All datasets used in our experiments are publicly available, with data processing procedures specified in Section 4.1.

## D  APPENDIX

### D.1  FAST INFERENCE ALGORITHM FOR COIN

The challenge of selecting the optimal subset of tokens is an NP-hard problem. Therefore, we use a fast greedy algorithm to efficiently find an optimal solution. The core of this acceleration lies in an incremental update method for the QR decomposition in each selection round. We select k tokens iteratively. In each round, we select the token that maximizes a hybrid score, which balances coverage and informativeness.

---

**Algorithm 1** Fast Greedy Selection Algorithm

---

**Require:** Tokens $\mathbf{X} \in \mathbb{R}^{N \times D}$, Informativeness scores $\mathbf{p} \in \mathbb{R}^{N \times 1}$, retained size $k$, hyperparameter $\Lambda$.
**Ensure:** A set of selected token indices $\mathcal{S}$.
1: $\mathcal{S} \leftarrow \emptyset, \mathcal{U} \leftarrow \{1, \ldots, N\}$
2: $\mathbf{C} \leftarrow \mathbf{0} \in \mathbb{R}^{N \times k}$ Cache for projection coefficients
3: $\mathbf{Q} \leftarrow \emptyset, s \leftarrow 0$ Orthonormal basis and its size
4: $j \leftarrow \arg\max_{j \in \mathcal{U}} p_j$
5: **for** $i = 1$ to $k$ **do**
6:     $\mathcal{S} \leftarrow \mathcal{S} \cup \{j\}$
7:     $\mathcal{U} \leftarrow \{1, \ldots, N\} \setminus \mathcal{S}$
8:     $\mathbf{x}_{new} \leftarrow \mathbf{X}_j$
9:     $\mathbf{q}_{new} \leftarrow \text{GramSchmidt}(\mathbf{x}_{new}, \mathbf{Q})$
10:     $\mathbf{Q} \leftarrow [\mathbf{Q} \mid \mathbf{q}_{new}^{\mathsf{T}}]$
11:     $s \leftarrow s + 1$
12:     $\mathbf{C}_{:,s} \leftarrow \mathbf{X} \cdot \mathbf{q}_{new}$
13:     **if** $i < k$ **then**
14:         $\mathbf{d}^2 \leftarrow 1 - \sum_{m=1}^s \mathbf{C}_{j,m}^2$ for $j \in \mathcal{U}$
15:         $\mathbf{S} \leftarrow \Lambda \cdot \mathbf{d}' + (1 - \Lambda) \cdot \mathbf{p}$ for $j \in \mathcal{U}$
16:         $j \leftarrow \arg\max_{j \in \mathcal{U}} \mathbf{S}_j$
17:     **end if**
18: **end for**
19: **return** $\mathcal{S}$

---

## D.2 BENCHMARKS

**GQA.** A large-scale visual question answering benchmark designed to evaluate compositional reasoning and visual understanding. It provides detailed question-answer pairs covering objects, attributes, and relationships. We follow the standard test-dev balanced split for evaluation.

**MMBench.** A comprehensive benchmark designed to evaluate the multi-modal understanding capabilities of large language models. It consists of a diverse set of multiple-choice questions that cover a wide range of tasks, from basic perception and object recognition to complex cognitive reasoning and world knowledge.

**MME.** A comprehensive benchmark measures a model's performance across 14 distinct subtasks, which are divided into two main categories: perception and cognition. Perception tasks test a model's ability to recognize and understand basic visual elements like objects, text, or a scene's context. In contrast, cognition tasks evaluate its higher-level reasoning skills, such as applying common sense, performing logical inference, or solving math and science problems based on an image. The MME benchmark is designed to provide a detailed and fair comparison of MLLMs by using carefully designed instruction-answer pairs and covering a wide range of domains to identify a model's strengths and weaknesses.

**POPE.** A benchmark designed to rigorously assess object hallucination in large vision-language models. It systematically presents a model with a series of "Yes or No" questions about the existence of specific objects in an image. By strategically sampling objects that are not present, POPE effectively measures a model's tendency to incorrectly confirm or generate objects, providing a robust method for evaluating a model's honesty and accuracy.

**TextVQA.** A benchmark dataset for Visual Question Answering (VQA) that requires models to answer questions by reading and reasoning about text within images. To succeed, a model must first perform accurate Optical Character Recognition (OCR) to extract the text and then combine this information with the visual context of the image. This makes TextVQA a crucial test for a model's ability to integrate visual perception with linguistic understanding.

**VizWiz.** A unique Visual Question Answering (VQA) dataset that focuses on questions asked by people who are visually impaired. Its images are often of poor quality, including blurriness, suboptimal framing, or occlusions, because they were captured by users seeking assistance. The questions are also grounded in real-world needs and are often much more open-ended. The purpose of the VizWiz benchmark is to push the development of VQA models that are robust to real-world visual imperfections and can provide practical, useful information to assist people with vision loss.

**OCRBench.** A comprehensive evaluation benchmark designed to assess the optical character recognition (OCR) capabilities of large multi-modal models. It tests a model's ability to handle a wide variety of tasks, including text localization, understanding handwritten content, and performing logical reasoning based on the text found in an image. The benchmark includes diverse scenarios such as receipts, documents, and street scenes to provide a robust evaluation of a model's visual and linguistic understanding in real-world, text-rich environments.

**ScienceQA.** A large-scale benchmark designed to evaluate multi-modal reasoning by presenting models with complex science questions. It includes not only images and text but also detailed rationales—step-by-step explanations for the correct answers. This feature allows researchers to assess a model's underlying reasoning process, rather than just its final output. The questions cover a diverse range of science topics from elementary to high school levels.

**RealWorldQA.** A benchmark designed to evaluate a multi-modal model's real-world spatial understanding and common sense reasoning. The dataset consists of high-resolution images, often captured from vehicles or other real-world scenarios, each paired with a question and a verifiable answer. Unlike many other benchmarks, RealWorldQA focuses on challenging models to recognize subtle details and perform complex reasoning based on their visual perception. This allows for

a robust assessment of a model's ability to comprehend our physical world and act as a practical assistant.

