# OpenReview forum: "CoIn: Coverage and Informativeness-Guided Token Reduction for Efficient Large Multimodal Models"
_ICLR.cc/2026/Conference — ICLR 2026 Conference Withdrawn Submission_

### Official Review · Reviewer_u9rW · 2025-10-29

**Soundness:** 2
**Presentation:** 3
**Contribution:** 2
**Rating:** 4
**Confidence:** 4

**Summary:**

This paper addresses the challenges of low inference efficiency and high memory consumption in large multimodal models (LMMs) caused by the excessive number of visual tokens. Unlike existing approaches based on importance, diversity, or two-stage hybrid strategies, this work introduces a token pruning method guided by two self-defined metrics—informativeness and coverage. Experiments across multiple benchmarks demonstrate that CoIn effectively reduces computational and memory costs while maintaining competitive performance.

**Strengths:**

1.	The paper proposes a training-free framework, CoIn, which unifies visual saliency, cross-modal semantic alignment, and representational diversity within a joint selection process to identify visual tokens with higher informativeness and coverage.
2.	Experimental results show that CoIn consistently outperforms other baselines across various visual token pruning ratios, achieving superior performance while maintaining efficiency.

**Weaknesses:**

1.	Although the paper introduces two new metrics to measure informativeness and coverage, it does not provide new insights compared with previous two-stage hybrid approaches, which also aim to balance importance and diversity. Moreover, while the method formulates token pruning as a subset selection problem, the adopted greedy strategy does not seem to fully exploit its potential. It would be interesting to explore heuristic search algorithms or adaptive pruning mechanisms that can dynamically determine when to stop pruning based on the relationship between the selected and remaining subsets—thereby achieving an optimal trade-off between performance and efficiency, rather than pruning by a fixed percentage.
2.	The model appears to be overly sensitive to hyperparameters. As shown in Tables 1, 2, and 3, the values of α and β vary across model sizes, architectures, and pruning ratios, suggesting that careful tuning is required for each benchmark. Furthermore, Section 4.5 shows that α and β are also dataset-sensitive, implying that it may be difficult to find a universally effective set of hyperparameters in real-world applications.
3.	Several descriptions lack clarity. For instance, in Section 3.2, the statement “we visualize token scores when using only intrinsic saliency or only text relevance” is ambiguous—are intrinsic saliency and text relevance derived from prior methods? In addition, the font size in Figure 2(c) is too small and should be enlarged for readability. Table 3 also contains a minor typo: CoIn’s score on RWQA should be corrected from 54.12 to 54.1.
4.	The paper claims state-of-the-art (SoTA) performance but lacks comparisons with several strong baselines, such as FastV [1] and DART [2], which are both competitive token reduction frameworks.

References
 [1] An Image is Worth 1/2 Tokens After Layer 2: Plug-and-Play Inference Acceleration for Large Vision-Language Models. ECCV 2024.
 [2] Stop Looking for “Important Tokens” in Multimodal Language Models: Duplication Matters More. EMNLP 2025.

**Questions:**

1.	How are the values of α and β selected across different tests? Are they determined based on the evaluation results?

---

### Official Review · Reviewer_MhGj · 2025-10-30

**Soundness:** 3
**Presentation:** 3
**Contribution:** 2
**Rating:** 4
**Confidence:** 5

**Summary:**

This paper presents CoIn, a training-free visual token reduction framework that accelerates Large Multimodal Models (LMMs) by jointly optimizing two principles: informativeness – captures a token’s visual saliency and text–visual semantic relevance and coverage – enforces diversity via QR-based orthogonal residuals, ensuring complementary visual representation. Unlike two-stage hybrids (e.g., PruMerge+ or VisionZip) that separately handle importance and diversity, CoIn unifies them into a single scoring function. It uses a fast greedy selection algorithm with incremental QR updates to select the most representative subset of visual tokens. CoIn is training-free, compatible with FlashAttention and KV-cache, and tested on LLaVA-1.5 (7B/13B) and LLaVA-NeXT-7B. It marginally outperforms DivPrune, VisionZip (CVPR 2025), and PruMerge+ under equivalent compression ratios. With 94.4 % visual tokens removed, CoIn retains ≈91 % task performance and reduces latency by 6.5× and GPU memory by ~15 % on LLaVA-NeXT.

**Strengths:**

1. Joint informativeness–coverage objective grounded in subspace theory.
2. Works with frozen pretrained LMMs; no retraining or fine-tuning needed.
3. Compatible with existing accelerators and inference frameworks.

**Weaknesses:**

1. The trade-off weights \alpha (coverage vs. informativeness) and \beta (visual vs. text cues) are manually tuned per dataset. Optimal values vary between reasoning-heavy and OCR-heavy tasks, limiting plug-and-play deployment.
2. Experiments are restricted to LLaVA-based models. No results on modern architectures such as Qwen-VL / Qwen2.5-VL, CogVLM, InternVL2, or Video-LLaVA.
3. CoIn does not evaluate on video-LMMs or temporal benchmarks (MSVD-QA, MSRVTT-QA, ActivityNet-QA). Its applicability to temporal or sequential visual reasoning remains untested.
4. The incremental QR selection has no formal optimality guarantee or complexity analysis for large token counts.

**Questions:**

1. Could \alpha and \beta be estimated adaptively (e.g., via entropy or attention dispersion) to remove dataset dependence?
2. Have you tested CoIn on newer models such as Qwen-VL 2.5, CogVLM, or InternVL2?
3. Can CoIn extend naturally to video or multi-image inputs?
4. What is the computational complexity and time overhead of the incremental QR step for 2,880 tokens?

---

### Official Review · Reviewer_LdWb · 2025-10-31

**Soundness:** 2
**Presentation:** 3
**Contribution:** 2
**Rating:** 2
**Confidence:** 4

**Summary:**

This paper presents ​​CoIn​​, a training-free token reduction framework for Large Multimodal Models. The authors identify limitations in existing token reduction methods and propose a unified optimal subset selection approach guided by two criteria: ​​informativeness​​ (combining visual saliency and text relevance) and ​​coverage​​ (diversity). Experimental results indicate that CoIn achieves superior performance over existing methods, though primarily under specific hyperparameter settings and when evaluated on CLIP-based LLaVA models. However, the generalizability of this approach appears limited, and its broader applicability requires further validation.

**Strengths:**

1.The paper is well-written with clear language.
2.The critique of existing methods and the reformulation of token reduction as a joint optimization problem are insightful.
3.The proposed method demonstrates certain effectiveness by outperforming existing token compression approaches under specific parameter configurations.

**Weaknesses:**

1.The method relies heavily on the hyperparameters α and β. This may limit practicality and generalization. While Figure 4 shows the impact of these parameters, the paper lacks clear heuristics for setting them without extensive tuning.
2.The text relevance component uses cosine similarity between visual and text tokens, which may be influenced by the specific design of the vision encoder and text encoder. The experiments are primarily based on CLIP, raising concerns about generalizability to other encoders (e.g., SigLIP).

**Questions:**

Can you provide better methods for hyperparameter tuning, as well as supplement with some experiments based on other image encoders?

---

### Official Review · Reviewer_7LzG · 2025-10-31

**Soundness:** 2
**Presentation:** 2
**Contribution:** 3
**Rating:** 6
**Confidence:** 2

**Summary:**

This paper addresses the computational bottleneck in Large Multimodal Models (LMMs) caused by the quadratic scaling of inference time and memory with respect to visual token count. The authors propose CoIn, a training-free framework for visual token reduction that formulates the problem as an optimal subset selection task guided by two key criteria: **informativeness** (combining visual saliency and text relevance) and **coverage** (ensuring diversity through representational novelty). Unlike existing methods that either focus solely on importance (leading to redundancy) or diversity (missing critical information), CoIn jointly optimizes both criteria through a unified scoring function. The method uses a greedy selection algorithm with QR decomposition to iteratively select tokens that maximize the combined score. Experiments on LLaVA models show that CoIn can reduce visual tokens by up to 94.4% while maintaining 91.29% of original performance across 9 benchmarks.

**Strengths:**

1. **Novel Problem Formulation**: The paper reframes token reduction as a principled optimization problem with clear theoretical motivation, moving beyond heuristic approaches to a more systematic framework.

2. **Strong Empirical Results**: CoIn consistently outperforms baselines across multiple models (LLaVA-1.5-7B/13B, LLaVA-NeXT) and benchmarks, showing particularly impressive performance at extreme compression ratios (94.4% reduction).

3. **Practical Design Choices**: The method is training-free, model-agnostic (no [CLS] token dependency), and compatible with modern inference optimizations like FlashAttention and KV caching, making it readily deployable.

4. **Comprehensive Evaluation**: The paper includes extensive experiments across 9 diverse benchmarks covering VQA, OCR, reasoning, and hallucination detection tasks, with detailed ablation studies validating each component.

5. **Efficient Implementation**: The use of incremental QR decomposition for coverage computation is computationally efficient, achieving 6.5× speedup in prefill time.

6. **Clear Presentation**: The paper effectively visualizes the limitations of existing methods (Figure 1) and provides intuitive explanations for the complementary nature of visual saliency and text relevance (Figure 2).

**Weaknesses:**

1. **Limited Theoretical Analysis**: While the formulation is principled, the paper lacks theoretical guarantees about the approximation quality of the greedy algorithm or bounds on performance degradation.

2. **Hyperparameter Sensitivity**: The method introduces two hyperparameters (α and β) that require tuning. While ablations are provided, optimal values appear task-dependent, potentially limiting the method's generalizability.

3. **Computational Overhead**: Despite efficiency claims, the QR decomposition adds computational cost during token selection. The paper doesn't compare selection time overhead against baselines.

4. **Limited Scope**:
   - Evaluation is restricted to CLIP-based vision encoders
   - Only static image understanding is tested (no video or multi-image scenarios)
   - No comparison with training-based pruning methods

5. **Incomplete Analysis**:
   - Missing analysis of which types of visual content are preferentially retained/discarded
   - No failure case analysis or discussion of when the method might underperform
   - Limited discussion of the relationship between image complexity and optimal retention rate

6. **Experimental Limitations**:
   - All experiments use batch size 1, which may not reflect real-world deployment scenarios
   - No evaluation on recent challenging benchmarks like MM-Vet or LLaVA-Bench

**Questions:**

1. **Scalability**: How does the computational cost of QR decomposition scale with the number of visual tokens? What is the break-even point where selection overhead outweighs inference savings?

2. **Generalization**: Have the authors tested CoIn with non-CLIP vision encoders (e.g., DINOv2, SAM) or different projection methods? How robust is the method to architectural variations?

3. **Dynamic Selection**: Could the retention ratio K be adaptively determined based on image complexity rather than being fixed? What signals could guide such adaptation?

4. **Token Importance Distribution**: Can the authors provide visualizations showing which image regions are consistently selected/discarded across different images and tasks?

5. **Failure Analysis**: In what scenarios does CoIn underperform baselines? Are there specific visual patterns or question types where the joint optimization fails?

6. **Theoretical Guarantees**: Can the authors provide any approximation guarantees for the greedy algorithm relative to the optimal solution of Equation 2?

7. **Cross-Modal Alignment**: How does the quality of vision-language alignment in the base model affect CoIn's performance? Would the method work with weakly aligned models?

8. **Extension to Other Modalities**: Could this framework be extended to video (temporal tokens) or audio-visual models? What modifications would be necessary?

---

### Note · Authors · 2025-11-12

I have read and agree with the venue's withdrawal policy on behalf of myself and my co-authors.